# Continuous Detection of Stimulus Brightness Differences Using Visual Evoked Potentials in Healthy Volunteers with Closed Eyes

**DOI:** 10.3390/bioengineering11060605

**Published:** 2024-06-13

**Authors:** Stephan Kalb, Carl Böck, Matthias Bolz, Christine Schlömmer, Lucija Kudumija, Martin W. Dünser, Jens Meier

**Affiliations:** 1Department of Anesthesiology and Intensive Care Medicine, Kepler University Hospital GmbH, Johannes Kepler University Linz, 4040 Linz, Austria; 2JKU Linz Institute of Technology SAL eSPML Lab, Institute of Signal Processing, Johannes Kepler University Linz, 4040 Linz, Austria; carl.boeck@jku.at; 3JKU Department of Ophthalmology, Kepler University Hospital GmbH, Johannes Kepler University Linz, 4040 Linz, Austria

**Keywords:** visual evoked potentials, chromatic, red light, green light, closed eyelids, machine learning

## Abstract

**Background/Objectives:** We defined the value of a machine learning algorithm to distinguish between the EEG response to no light or any light stimulations, and between light stimulations with different brightnesses in awake volunteers with closed eyelids. This new method utilizing EEG analysis is visionary in the understanding of visual signal processing and will facilitate the deepening of our knowledge concerning anesthetic research. **Methods**: X-gradient boosting models were used to classify the cortical response to visual stimulation (no light vs. light stimulations and two lights with different brightnesses). For each of the two classifications, three scenarios were tested: training and prediction in all participants (all), training and prediction in one participant (individual), and training across all but one participant with prediction performed in the participant left out (one out). **Results**: Ninety-four Caucasian adults were included. The machine learning algorithm had a very high predictive value and accuracy in differentiating between no light and any light stimulations (AUCROC_all_: 0.96; accuracy_all_: 0.94; AUCROC_individual_: 0.96 ± 0.05, accuracy_individual_: 0.94 ± 0.05; AUCROC_oneout_: 0.98 ± 0.04; accuracy_oneout_: 0.96 ± 0.04). The machine learning algorithm was highly predictive and accurate in distinguishing between light stimulations with different brightnesses (AUCROC_all_: 0.97; accuracy_all_: 0.91; AUCROC_individual_: 0.98 ± 0.04, accuracy_individual_: 0.96 ± 0.04; AUCROC_oneout_: 0.96 ± 0.05; accuracy_oneout_: 0.93 ± 0.06). The predictive value and accuracy of both classification tasks was comparable between males and females. **Conclusions**: Machine learning algorithms could almost continuously and reliably differentiate between the cortical EEG responses to no light or light stimulations using visual evoked potentials in awake female and male volunteers with eyes closed. Our findings may open new possibilities for the use of visual evoked potentials in the clinical and intraoperative setting.

## 1. Introduction

Visual evoked potentials (VEPs) record the EEG signals generated in the occipital cortex in response to light stimulations of the retina. Thus, VEPs can assess the integrity of the retina, optic nerve, visual pathways, and occipital cortex. Although mostly used in conscious patients, VEPs have also been recorded under general anesthesia with the shortcoming of lid closure, lowering the quality of its measurements. For example, flash VEPs are used in the intraoperative setting as a monitoring tool during neurosurgical procedures, prolonged surgeries in the prone position, or as an indirect marker of intracranial hypertension [1].

However, the use of current VEP technology in the intraoperative setting is challenged by relevant shortcomings [2]. When analyzing VEPs, there is no universally agreed standard to define significant changes in the EEG signal. Typically, changes in the latency of selected wave peaks >1 milliseconds or amplitude reductions >50% are used to define deviations from normality or previous measurements [1]. The detection of such subtle changes in a noisy signal, such as the EEG, is difficult and requires averaging hundreds of sweeps to identify changes in the VEP morphology between different clinical situations (e.g., during neurosurgery adjacent to the optic pathways). Since each measuring process takes several minutes and interpretation usually requires a neurophysiologist, continuous monitoring and timely detection of such changes is nearly impossible.

In a proof-of-concept study, our working group found that an artificial neuronal network with one hidden layer could distinguish cortical color perception by analyzing a low number of chromatic VEPs in healthy volunteers visually stimulated by flickering red/black or green/black checkerboards [3]. For this classification task, only nine stimulations were needed to find subtle differences between red and green VEPs. The classification accuracy was highest if the model was trained on the individual subject. 

The aim of the present study was to improve this approach by verifying its feasibility in healthy volunteers with closed eyelids.

## 2. Materials and Methods

This study was designed as an experimental healthy volunteer study, conducted at the Johannes Kepler University in Linz, Austria. Experiments were performed during the time from November 2021 until May 2022. The study protocol was reviewed and approved by the ethics committee of the Johannes Kepler University (protocol code 1201/2020, date of approval: 11 April 2020). Written informed consent was obtained from all participants before study enrolment. 

### 2.1. Study Participants

Subjects aged between 18 and 75 years were eligible for inclusion in this study. Severe systemic disease (defined as American Society of Anesthesiologists physical status classification system of III or higher), a history of seizures, an ophthalmological or neurological disease, red–green color vision deficiency, and claustrophobia were the exclusion criteria. Before enrolment, all subjects underwent testing for red–green color vision deficiency using standard Ishihara plates [4,5]. A cut-off value of ≥10 correct results out of 12 test plates was used to exclude red–green color vision deficiency.

### 2.2. Experimental Set-Up (Figure 1)

All experiments were conducted in a quiet, darkened room. During the experiments, only the study participant and one researcher were present in the room. Study participants were comfortably placed in a semi-recumbent position on an examination bed and were awake but had their eyelids closed during the entire period of the experiment.

**Figure 1 bioengineering-11-00605-f001:**
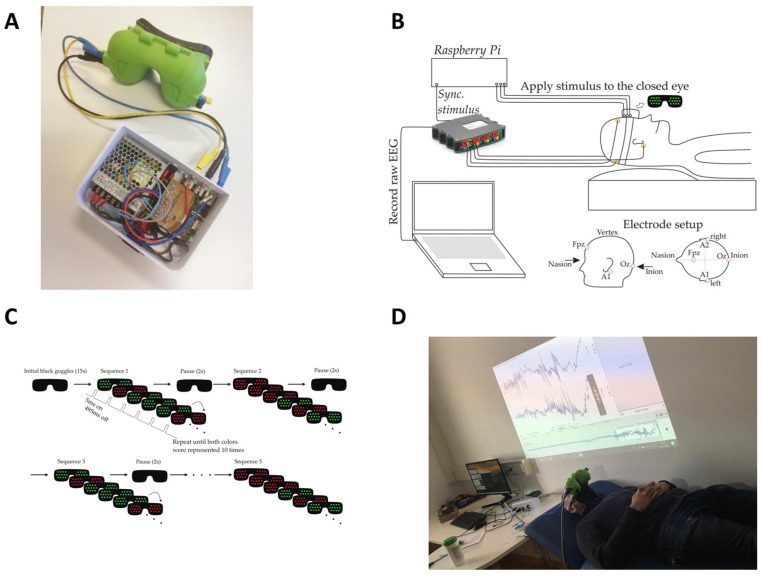
LED googles connected to the Raspberry Pi (**A**), schematic experimental set-up (**B**), stimulation sequence (**C**), and picture of experiment (**D**).

Special LED goggles were developed for this experiment (Figure 1A). They completely enclosed the orbita so that no ambient light could reach the eyes and interfere with VEP stimulation. During the VEP stimulation, LEDs were emitting green and red light, which was used for a visual stimulation through closed eyelids. The frame of the 3D-printed goggles was made of an FDA-approved biopolymer, which was rimmed by skin-compatible, two-component silicone to ensure absolute darkening and elimination of ambient light pollution. Eight light-emitting diodes with a maximum illuminance of 20,000 lux and a variable stimulation frequency of 1–15 Hz were installed inside the goggles, overlying each eye. The luminance intensity with which green and red light was emitted was calibrated to a value of 1000 lux. A Raspberry Pi single-board computer was programmed to control the goggles’ stimulation patterns and colors. Gold cup electrodes were mounted over the mid occipital (OZ) and frontal (Fpz) scalp after roughening of the skin with a prepping gel to achieve a skin impedance < 5 kΩ. Further electrodes were placed behind the ear lobes on the left (A1) and right (A2) sides, with the latter serving as the ground electrode. Cortical responses to visual stimulations were recorded over the occipital (OZ against Fpz) lobe using a high-performance and high-accuracy biosignal amplifier (g.USBamp Research; g.tec medical engineering GmbH, Schiedlberg, Austria) at a sampling frequency of 1200 Hz and a bit resolution of 24 bit. In order to obtain the highest possible accuracy for measuring the cortical response, both the Raspberry Pi single-board computer and the biomedical signal amplifier were synchronized by using an external trigger cable (Figure 1B). The time instants of applying the stimulus (light or no light, red or green light) to the eye were recorded by the biomedical signal amplifier, and the raw signal (in the range of ±20 mV) was sliced accordingly.

### 2.3. Study Experiment

The experiment consisted of 10 cycles. Each cycle lasted 3 min and 45 s, during which red or green flashes were randomly applied at a frequency of 2 Hz (up to a total of 1500 flashes) to the closed eyes of the study participants. After every cycle, the well-being of the study participants was checked, and subjects were allowed to move if necessary.

### 2.4. Study Goals

As the primary study goals, we aimed to define the value of a machine learning algorithm to distinguish (1) between the cortical EEG response to no light or light stimulations, and (2) between the cortical EEG response to light stimulations with different brightnesses using red or green light stimulations in awake volunteers with closed eyelids. The main outcome parameters were the area under the receiver operator characteristic curve (AUCROC) and accuracy of the algorithm. Evaluation of the predictive value of single signal components as well as the influence of sex on the predictive value of the algorithm were secondary study goals.

### 2.5. Data Processing and Statistical Analysis

All raw EEG signals collected during the experiment were stored as MatLab files on a notebook. Pre-processing of all recorded data was conducted as follows: A 50 Hz notch filter was applied to suppress power line interference. A bandpass filter with cut-off frequencies of 0.1 Hz and 100 Hz was used to eliminate both baseline drifts and high-frequency noise. Subsequently, the recordings were sliced and aligned based on the recorded switching points (initiation of every single flash), which were captured using an electrical signal from the Raspberry Pi computer (Figure 1). We received 3000 single trials of 500 ms length per subject, 1500 for red light stimulations and 1500 for green light stimulations. Based on the methodology used in our proof-of-concept study [3], single sweeps were averaged across ten stimulations to increase signal quality. Nine-hundred sweeps, during which no stimulation took place, were taken in every subject as well. Consequently, we used signal slices of 500 ms (averaged across 10 sweeps) for classification.

In order to identify the most appropriate machine learning algorithm to answer the primary study question, we performed the classification task to predict the cortical EEG response to no light or light stimulations in all study participants using five different machine learning algorithms (X-gradient boosting, random forests, neuronal networks, adaBoost, general linear models). The algorithm exhibiting the highest AUCROC was subsequently used to answer the primary and secondary study questions. The median and 95% confidence intervals of the AUCROC of the different machine learning algorithms to predict the cortical EEG response to no light or light stimulations in all study participants were as follows: X-gradient boosting, 0.91 (0.9–0.91); neuronal networks, 0.89 (0.88–0.91); general linear models, 0.89 (0.88–0.9); random forests, 0.87 (0.86–0.88); adaBoost, 0.76 (0.75–0.78).

X-gradient boosting models were hence used (1) to classify the EEG response to no light or light stimulations, and (2) to classify the EEG response to red or green light stimulation. For each of the two classifications, we tested three different scenarios: training and prediction across all participants (all), training and prediction in an individual participant (individual), and training across all but one participant with prediction performed in the participant left out of the training set (one out) [6]. For the first two scenarios (all, individual), we randomly split the cohort of measurements into a training, validation, and test dataset. The training and validation datasets were used for model training (80% and 10% of data, respectively), whereas the test dataset (10% of data) was used to test the algorithm. For the third scenario (one out), the training and test datasets were not randomly selected but chosen according to the subjects. Accordingly, we carried out a 94-fold cross-validation using 93 subjects as the training set and one subject as the independent test set. The AUCROC and accuracy were calculated for all measurements performed in the test datasets.

## 3. Results

Ninety-four Caucasian subjects [male sex: 41 (43.6%); age, 28 ± 12 years; body mass index, 23.4 ± 3.5 kg/m²] were included in the experiment. All participants completed the entire experimental protocol. No technical errors or adverse events occurred. There were no missing values in the dataset. All participants tolerated the stimulation well, and no measurements had to be interrupted due to physical problems.

The machine learning algorithm had a very high predictive value (AUCROC) and accuracy to correctly classify the cortical response between no light stimulations and light stimulations in all three scenarios (Figure 2). When all participants were analyzed (all), the AUCROC was 0.96 and the accuracy was 0.94. For the prediction of the cortical response of an individual participant (individual), the AUCROC was 0.96 ± 0.05 and the accuracy was 0.94 ± 0.05. Finally, the AUCROC was 0.98 ± 0.04 and the accuracy was 0.96 ± 0.04 when the EEG response of individuals who were not included in the training and validation set was predicted (one out). Similarly, the machine learning algorithm was highly predictive and accurate in correctly classifying the cortical response to light stimulations with different brightnesses (green or red color stimulations) (AUCROC_all_: 0.97; accuracy_all_: 0.91; AUCROC_individual_: 0.98 ± 0.04, accuracy_individual_: 0.96 ± 0.04; AUCROC_oneout_: 0.96 ± 0.05; accuracy_oneout_: 0.93 ± 0.06, respectively) (Figure 3). 

The predictive value (AUCROC) and accuracy of both classification tasks was comparable between the male and female participants (Figure 4). All components of the EEG signal were significant features informing the machine learning algorithm.

## 4. Discussion

In this experimental study, machine learning algorithms achieved an excellent predictive value and a very high accuracy for detecting VEPs in the EEG signal and for distinguishing between the VEP responses to no light or light stimulations as well as to light stimulations with different brightnesses in awake volunteers with closed eyelids. Although the AUCROC was highest when algorithms were trained on the same individual in whom the prediction was made, the algorithms’ predictive values and accuracies remained very high when they were trained on 93 study participants and applied to one individual not included in the training and validation set.

As this was an experimental set-up, important differences to the clinical setting need to be mentioned. While notch filters are, for example, commonly used in experimental settings, their use in clinical practice is not recommended as they may reduce the EEG signal [7]. Similar to our proof-of-concept study, we chose a machine learning method to build the prediction algorithms. However, unlike in the previous study, in which a single layer neuronal network was applied [3], we used the tree-based X-gradient boosting model to analyze the raw EEG data in the present study, as this algorithm showed the highest predictive value to answer the main classification task in this dataset. X-gradient boosting models are effective in analyzing biosignals and allow to build simpler algorithms from complex raw signals than neuronal networks. Similarly, when comparing the predictive value and accuracy of the machine learning algorithm in this study to the algorithms reported in our previous study, the X-gradient boosting model showed better prediction and accuracy than the neuronal network when classifying the VEP response to light stimulations with different brightnesses (green- or red-light stimulations) in one individual when the algorithm was trained on the remaining study population [3].

All components of the cortical EEG signal were important to inform the machine learning algorithm. Future work needs to elucidate whether feature reductions of the raw dataset, for example, by using a principal component or wavelet analysis, could allow to focus on fewer components of the raw signal and therefore permit to build more complex machine learning algorithms with even better classification abilities while not prolonging analysis cycle times.

Three results of our study are novel and deserve to be highlighted when comparing this experiment to previous research on chromatic VEPs [8,9,10,11]. Firstly, and similar to our proof-of-concept study, the algorithm applied in the present experimental set-up could reliably classify the EEG response to visual stimulation using only very few VEPs (<10 VEPs at 2 Hz). This allowed for an accurate VEP interpretation at intervals as short as <5 s. From a clinical perspective, this appears to be the most important result and strength of our study. In contrast to standard, intermittent VEP interpretations by either neurophysiologists or other machine learning models [12], the algorithm allows for a nearly continuous VEP analysis. Possible clinical scenarios in which rapid or continuous VEP interpretations would be helpful comprise, for example, a VEP analysis in infants or very young children, patients with severe trauma unable to respond or attend to a stimulus for long periods of time, and a real-time VEP analysis during surgery. 

Second, the machine learning algorithms in this study could not only distinguish between no light stimulations or light stimulations but could also reliably classify the cortical VEP response to light stimulations with different brightnesses. Although the light emitting diodes were calibrated to emit green and red light at the same luminance intensity, the eyelid served as a color filter as the estimated light transmission through the eyelid was found to be 0.3% for green light and 5.6% for red light [13]. Therefore, the cortical EEG responses detected in our study corresponded to light stimulations with different brightnesses and not stimulations with different colors. The ability of machine learning algorithms to correctly differentiate between VEP responses both in a binary and semi-qualitative fashion is, however, likely to increase its sensitivity to detect even subtle functional changes in the optic pathway. Accordingly, chromatic VEPs in patients with open eyes were reported to be more sensitive than standard VEPs when detecting color vision deficiencies in infants [5], short-term variations in blood glucose levels in diabetic patients [14], various forms of optic neuropathy [8,9], acquired deficiencies in color vision capacity [15], and early visual disturbances in patients with Parkinson’s disease [10].

Third, visual stimulations in our experiment were performed when study subjects had their eyelids closed. This is an essential difference to standard VEP practice in which patients are examined with their eyelids being open, which makes this technique particularly interesting for intraoperative use in anesthetized patients when, for example, accurate monitoring of the integrity of the optic pathways during neurosurgery [16] or prolonged prone positioning [17] is warranted. Although the current techniques, such as the SightSafer^TM^ visual stimulator (Chippenham, UK), have been successfully used during surgery in anaesthetized patients, they still require the use of an intermittent VEP waveform analysis by experienced neurophysiologists [17,18]. Another future application of our algorithm could be to determine the depth of anesthesia [19]. Importantly, the influences of different anesthetics on the cortical VEP responses [18,19,20] and the algorithm’s performance need to be evaluated before.were evaluated before.

The sex of the study participants influenced neither the predictive value nor the accuracy of the machine learning algorithms. As our analysis included only young, healthy volunteers, the results of our analysis cannot be extrapolated to elderly patients [21,22] and those with relevant comorbidities. As surgical procedures are commonly performed in older adults and patients suffering from relevant comorbidities [21], future experiments need to test the performance of our algorithm in these populations as well. In addition, our study cohort did not include subjects with a red–green color vision deficiency, a common pathology among Caucasian men, with a prevalence of up to 8% [23]. In the future, the presence of the red–green color vision deficiency needs to be elucidated to determine whether the predictive value of these machine learning-based algorithms is being influenced.

Further limitations need to be considered when interpreting the findings of our study. Although we used standard anatomical landmarks for electrode placement [24] and assured a low skin impedance before the measurements were taken, we did not investigate whether further quality assurance steps for electrode mounting are necessary to achieve a better raw EEG signal for analysis. In addition, we used green and red light to simulate light stimulations with different brightnesses in our study. This was the result of pilot tests analyzing which light stimulations resulted in cortical VEP responses that were most different from each other. We can, therefore, not exclude that using only green or only red light with different brightnesses or another color with different brightnesses might have resulted in an even better classification ability of the machine learning algorithms.

## 5. Conclusions

In conclusion, machine learning algorithms could almost continuously and reliably differentiate between the cortical EEG responses to no light or light stimulations using VEPs in awake male and female volunteers when their eyes were closed. Our findings may open new possibilities for the use of VEPs in the clinical and intraoperative setting.

## Figures and Tables

**Figure 2 bioengineering-11-00605-f002:**
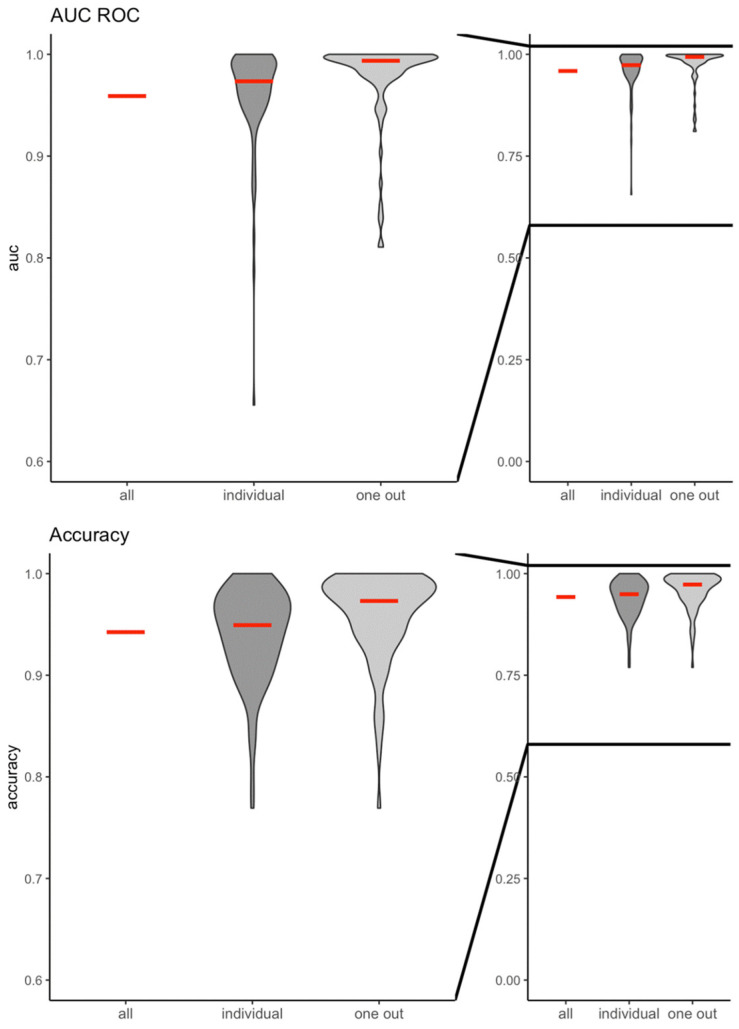
Violin plots of the area under the receiver operator characteristic curve (AUCROC) and accuracy for the prediction models including the entire study population (all), individual participants (individual), and all participants except one (one out) following no light or light stimulations.

**Figure 3 bioengineering-11-00605-f003:**
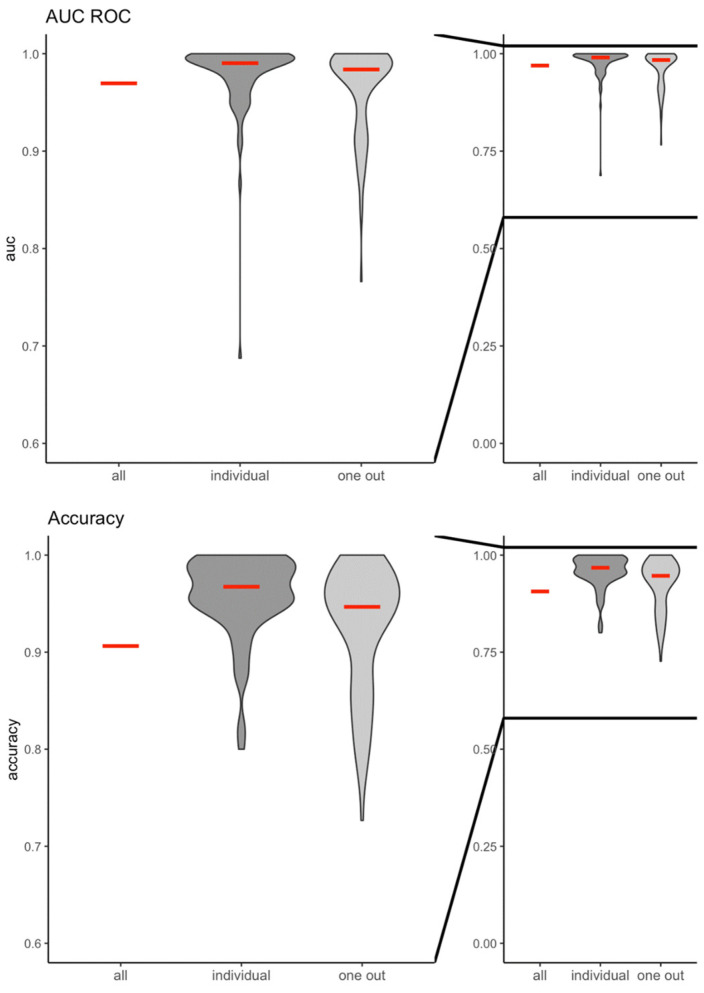
Violin plots of the area under the receiver operator characteristic curve (AUC ROC) and accuracy for the prediction model including the entire study population (all), individual participants (individual), and all participants except one (one out) following light stimulations with different brightnesses (green- or red-light stimulations).

**Figure 4 bioengineering-11-00605-f004:**
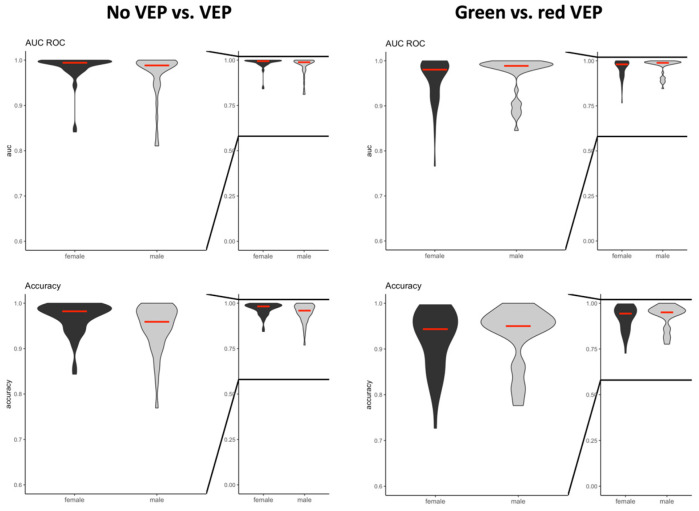
Violin plots of the area under the receiver operator characteristic curve (AUC ROC) and accuracy for both prediction models including the entire study population in females and males following no or any light stimulations, as well as following light stimulations with different brightnesses (green- or red-light stimulations).

## Data Availability

Anonymized data not published within this article will be made available upon request form any qualified investigator.

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
