# Peer review of "Continuous Detection of Stimulus Brightness Differences Using Visual Evoked Potentials in Healthy Volunteers with Closed Eyes"

_bioengineering, 2024, doi:10.3390/bioengineering11060605_

Round 1

Reviewer 1 Report

Comments and Suggestions for Authors

1. The summary section does not fully discuss the current similar studies.

2. A more detailed description of the experimental processes will help to understand the work. Add some photos of the experimental setup, as well as information about voltage signal values, acquisition timing, bit resolution of the measurements, inference time, etc.

3. The result part is too simple, please elaborate on the picture results.

4. Lack of comparison with other algorithms.

Comments on the Quality of English Language

The contents of the manuscript are easy to read and understand.

Author Response

We thank the reviewers for their insightful comments which helped to improve the quality of our manuscript.

  1. The summary section does not fully discuss the current similar studies.

Authors’ Response: In the revised manuscript, we have discussed and added further similar studies on this topic as suggested by reviewer 2. The respective references, which we have included, are:

  1. Uribe, A.A.; Mendel, E.; Peters, Z.A.; Shneker, B.F.; Abdel-Rasoul, M.; Bergesse, S.D. Comparison of visual evoked potential monitoring during spine surgeries under total intravenous anesthesia versus balanced general anesthesia. Clin. Neurophysiol. 2017, 128, 2006-2013, doi:10.1016/j.clinph.2017.07.420.
  2. Tanaka, R.; Tanaka, S.; Ichino, T.; Ishida, T.; Fuseya, S.; Kawamata, M. Differential effects of sevoflurane and propofol on an electroretinogram and visual evoked potentials. J. Anaesth. 2020, 34, 298-302, doi:10.1007/s00540-020-02733-7.

  1. A more detailed description of the experimental processes will help to understand the work. Add some photos of the experimental setup, as well as information about voltage signal values, acquisition timing, bit resolution of the measurements, inference time, etc.

Authors’ Response: The experimental set-up and processes were explained in more detail in the Materials and Methods section of the revised manuscript. A photo of the LED goggles connected with the Raspberry Pi (Figure 1A) and a photo of the experimental set-up while a participant underwent testing (Figure 1D) have been added. The requested information about the experimental process has also been introduced into the Materials and Methods section. The revised part of the manuscript now reads as follows: “Cortical responses to visual stimulation were recorded over the occipital (OZ against Fpz) lobe using a high-performance and high-accuracy biosignal amplifier (g.USBamp Research; g.tec medical engineering GmbH, Schiedlberg, Austria) at a sampling frequency of 1,200 Hz and a bit resolution of 24 bit. In order to obtain the highest possible accuracy for measuring the cortical response, both the Raspberry Pi single-board computer and the biomedical signal amplifier were synchronized by using an external trigger cable (Figure 1B). The time instants of applying the stimulus (light or no light, red or green light) to the eye were recorded by the biomedical signal amplifier, and the raw signal (in the range of ±20 mV) was sliced accordingly.”

  1. The result part is too simple, please elaborate on the picture results.

Authors’ Response: The Results section of the revised manuscript has been revised accordingly. We paid specific attention to better elaborate on the results shown in Figure 2, 3 and 4. For better understanding, we did this for Figure 2 in detail in order to set an example for interpretation of Figures 3 and 4. The respective part of the revised Results section reads as follows: “The machine learning algorithm had a very high predictive value (AUCROC) and accuracy to correctly classify the cortical response in differentiating between no light stimulation and light stimulation in all three scenarios (Figure 2). When all participants were analysed (all), the AUCROC was 0.96 and the accuracy 0.94. For prediction of the cortical response of an individual participant (individual) the AUCROC was 0.96±0.05, and the accuracy 0.94±0.05. Finally, the AUCROC was 0.98±0.04 and the accuracy 0.96±0.04 when the EEG response of individuals who had not been included in the training and validation set was predicted (one out).”

  1. Lack of comparison with other algorithms.

Authors’ Response: Thank you very much for this comment. In order to identify the optimum machine learning algorithm for model development, we have calculated the main classification task to differentiate the EEG response to light or no light stimulation in all study patients using five different machine learning algorithms (X-gradient boosting, random forests, neuronal networks, adaBoost, general linear models). Since the X-gradient boosting algorithm exhibited the highest predictive value, as assessed by the area under the receiver operating characteristic curve, this algorithm was chosen for model development. This information and the results were introduced into the Materials and Methods of the revised manuscript. The paragraph reads as follows: “In order to identify the most appropriate machine learning algorithm to answer the primary study question, we performed the classification task to predict the cortical EEG response to no light or light stimulation in all study participants using five different machine learning algorithms (X-gradient boosting, random forests, neuronal networks, adaBoost, general linear models). The algorithm exhibiting the highest AUCROC was subsequently used to answer the primary and secondary study questions. The median and 95% confidence intervals of the AUCROC of the different machine learning algorithms to predict the cortical EEG response to no light or light stimulation in all study participants were as follows: X-gradient boosting, 0.91 (0.9-0.91); neuronal networks, 0.89 (0.88-0.91); general linear models, 0.89 (0.88-0.9); random forests, 0.87 (0.86-0.88); adaBoost, 0.76 (0.75-0.78).”

In addition, the results were discussed in more detail in the revised Discussion section. The respective part of the revised Discussion section reads as follows: “However, unlike in the previous study, in which a single layer neuronal network was applied [3], we used the tree-based X-gradient boosting model to analyse the raw EEG data in the present study, as this algorithm showed the highest predictive value to answer the classification task in this dataset. X-gradient boosting models are effective in analysing biosignals and allow to build simpler algorithms from complex raw signals than neuronal networks. Similarly, when comparing the predictive value and accuracy of machine learning algorithm in this to the ones reported in our previous study, the X-gradient boosting model showed better prediction and accuracy than the neuronal network when classifying the VEP response to green or red colour stimulation in one individual when the algorithm was trained on the remaining study population [3].”

se of VEPs in the clinical and intraoperative setting.”

Reviewer 2 Report

Comments and Suggestions for Authors

This manuscript describes using machine learning algorithms (X boosting) to quickly characterize VEPs in subjects with closed eyes.  The authors claim their method was able to efficiently assess subjects’ detection of light vs no light stimulus conditions as well as red vs green light stimulus conditions.  This manuscript and a previous published pilot study focuses on the potential use of this technique in intraoperative settings. No evidence for why red/green discrimination might be a better way to monitor anesthesia depth is offered.  

The use of machine learning algorithms such as in this paper hold promise for more sensitive and efficient ways to assess the visual system using VEPS.  While the use of these machine learning algorithms shows promise for clinical applications,  there are serious problems with the conclusions drawn based on the experimental design.  I will argue their method did not allow for detection of “red” vs “green” lights.

Other Background

First it is worth noting that there are standards for clinical VEPs, none of which are referenced in the paper.  The use of a notch filter, while commonly done in research settings its use is generally thought to not be best practice in a clinical setting because it is near the optimum frequency for VEPs.  Other techniques to reduce noise are preferred.  If the 50 Hz noise cannot be reduced in any other way, it should at least be discussed in the methods and conclusions.

See for example:

Celesia G. G., Bodis-Wollner I., Chatrian G. E., Harding G. F. A., Sokol S., Spekreijse H. Recommended standards for electroretinograms and visual evoked potentials. Report of an IFCN committee. Electroencephalography and Clinical Neurophysiology. 1993;87(6):421–436..

Odom J. V., Bach M., Brigell M., et al. ISCEV standard for clinical visual evoked potentials (2009 update) Documenta Ophthalmologica. 2010;120(1):111–119. doi: 10.1007/s10633-009-9195-4

Odom, J.V., Bach, M., Brigell, M. et al. ISCEV standard for clinical visual evoked potentials: (2016 update). Documenta Ophthalmologica 133, 1–9 (2016). 

Other methods for intraoperative monitoring of VEPS such as Sight Saver are used in many and perform well.  While not as fast they work well in many situations encountered with surgeries.

e.g. Uribe, A. A., Mendel, E., Peters, Z. A., Shneker, B. F., Abdel-Rasoul, M., & Bergese, S. D. (2017). Comparison of visual evoked potential monitoring during spine surgeries under total intravenous anesthesia versus balanced general anesthesia. Clinical Neurophysiology128(10), 2006-2013.

The use of VEPs for measuring depth of anesthesia has been discussed in other papers not referenced for example: 

Tanaka, R., Tanaka, S., Ichino, T., Ishida, T., Fuseya, S., & Kawamata, M. (2020). Differential effects of sevoflurane and propofol on an electroretinogram and visual evoked potentials. Journal of anesthesia34, 298-302.  

Clearly some sedatives/anesthesia that are used (chloryl hydrate, propafol) have less impact on VEPs than other anesthesia such as sevoflurane.

Methods

The EEG system was not fully described.  What is the sampling rate of the system?  Was a minimum analysis time specified for the machine learning algorithms (250 msec is recommended in the Clinical Standards papers)?

Methodogical Error

A serious methodological error for this paper is not taking into account the spectral transmission of light through the eyelid.  There is significantly more longer wavelengths transmitted than other wavelengths, see for example.

Transmission through eyelid: Ando K, Kripke DF. Light attenuation by the human eyelid. Biol Psychiatry. 1996 Jan 1;39(1):22-5. doi: 10.1016/0006-3223(95)00109-3. PMID: 8719122.

This means that while the machine learning algorithms were picking up differences in detection for the “red” and “green” lights, in reality it was more likely detecting difference in differences in brightness of the two lights (mostly long wavelengths) filtered through the lid.  Even if the eyes were open the authors made no attempt to match the luminous efficiency (http://www.cvrl.org/lumindex.htm) of the different LEDSs to make them equiluminant. Photometric luminance matching of the two lights is necessary if any differences in detection of “red” vs “green” are discussed.

It would be very difficult to photometrically match a “red” and “green” light without measuring an individual’s eyelid transmission, this seems to weaken any discussion in the context of anesthesia and closed eyelids.  I think the paper would be stronger without any discussion of the red/green lights.

Despite not taking into account brightness rather than just color of their light stimuli, I think there is a strength in this paper.  The strength of this paper is its use of the machine learning algorithms to quickly differentiate responses in subjects. It is also worth noting that the X boosting algorithms differentiated better than neural networks.  This would be most useful in clinical care when speed and efficiency are critical, for example in getting VEP data from infants or very young children. Other applications in cases of patients with severe trauma and might not be able to respond or attend to a stimulus for long periods of time.

Author Response

We thank the reviewers for their insightful comments which helped to improve the quality of our manuscript.

The use of machine learning algorithms such as in this paper hold promise for more sensitive and efficient ways to assess the visual system using VEPS.  While the use of these machine learning algorithms shows promise for clinical applications, there are serious problems with the conclusions drawn based on the experimental design.  I will argue their method did not allow for detection of “red” vs “green” lights.

Authors’ Response: Please find our responses to the reviewer’s comment below.

First it is worth noting that there are standards for clinical VEPs, none of which are referenced in the paper.  The use of a notch filter, while commonly done in research settings its use is generally thought to not be best practice in a clinical setting because it is near the optimum frequency for VEPs.  Other techniques to reduce noise are preferred.  If the 50 Hz noise cannot be reduced in any other way, it should at least be discussed in the methods and conclusions.

See for example:

Celesia G. G., Bodis-Wollner I., Chatrian G. E., Harding G. F. A., Sokol S., Spekreijse H. Recommended standards for electroretinograms and visual evoked potentials. Report of an IFCN committee. Electroencephalography and Clinical Neurophysiology. 1993;87(6):421–436..

Odom J. V., Bach M., Brigell M., et al. ISCEV standard for clinical visual evoked potentials (2009 update) Documenta Ophthalmologica. 2010;120(1):111–119. doi: 10.1007/s10633-009-9195-4

Odom, J.V., Bach, M., Brigell, M. et al. ISCEV standard for clinical visual evoked potentials:

(2016 update). Documenta Ophthalmologica 133, 1–9 (2016). 

Authors’ Response: We used a 50 Hz notch filter in order to suppress power line interference. A bandpass filter with cut-off frequencies of 0.1 Hz and 100 Hz was applied to eliminate both baseline drifts and high frequency noise. This information is given more clearly in the revised manuscript. The fact that while notch filters are commonly used in experimental settings, they are not recommended by current ISCEV standards is highlighted and referenced (thank you!) in the Discussion section of the revised manuscript. The respective part of the revised Discussion section reads as follows: “As this was an experimental set-up, important differences to the clinical setting need to be mentioned. While notch filters are, for example, commonly used in experimental settings, their use in clinical practice is not recommended as they may reduce the EEG signal [7].”

Other methods for intraoperative monitoring of VEPS such as Sight Saver are used in many and perform well.  While not as fast they work well in many situations encountered with surgeries.

e.g. Uribe, A. A., Mendel, E., Peters, Z. A., Shneker, B. F., Abdel-Rasoul, M., & Bergese, S. D. (2017). Comparison of visual evoked potential monitoring during spine surgeries under total intravenous anesthesia versus balanced general anesthesia. Clinical Neurophysiology128(10), 2006-2013.

Authors’ Response: The suggested intraoperative technique to monitor VEPs is discussed and referenced in the Discussion section of the revised manuscript. The respective part reads as follows: “Although current techniques such as the SightSaferTM visual stimulator have been successfully used during surgery in the anaesthetized patient, they still require intermittent VEP waveform analysis by experienced neurophysiologists [17].”

The use of VEPs for measuring depth of anesthesia has been discussed in other papers not referenced for example: 

Tanaka, R., Tanaka, S., Ichino, T., Ishida, T., Fuseya, S., & Kawamata, M. (2020). Differential effects of sevoflurane and propofol on an electroretinogram and visual evoked potentials. Journal of anesthesia34, 298-302.  

Clearly some sedatives/anesthesia that are used (chloryl hydrate, propafol) have less impact on VEPs than other anesthesia such as sevoflurane.

Authors’ Response: Thank you very much for raising these important aspects. We have added them (including the references suggested) to the Discussion section of the revised manuscript. The respective part reads as follows: “Another possible application of our algorithm could be to determine the depth of anaesthesia [19]. Importantly, the influences of different anaesthetics on the cortical VEP responses [18-20] and the algorithm’s performance need to be evaluated before.”

Methods

The EEG system was not fully described.  What is the sampling rate of the system?  Was a minimum analysis time specified for the machine learning algorithms (250 msec is recommended in the Clinical Standards papers)?

Authors’ Response: In the Materials and Methods section of the revised manuscript, we have added further details on the experimental setup used, including the EEG system. The respective paragraph now reads as follows: “We received 3,000 single trials of 500 ms length per subject, 1,500 for red light stimulation and 1,500 for green light stimulation. Based on the methodology used in our proof-of-concept study [3], single sweeps were averaged across ten stimulations to increase the signal quality. Nine-hundred sweeps, during which no stimulation took place, were taken in every subject as well. Consequently, we used signal slices of 500 ms (averaged across 10 sweeps) for classification.”

Methodogical Error

A serious methodological error for this paper is not taking into account the spectral transmission of light through the eyelid.  There is significantly more longer wavelengths transmitted than other wavelengths, see for example.

Transmission through eyelid: Ando K, Kripke DF. Light attenuation by the human eyelid. Biol Psychiatry. 1996 Jan 1;39(1):22-5. doi: 10.1016/0006-3223(95)00109-3. PMID: 8719122.

This means that while the machine learning algorithms were picking up differences in detection for the “red” and “green” lights, in reality it was more likely detecting difference in differences in brightness of the two lights (mostly long wavelengths) filtered through the lid.  Even if the eyes were open the authors made no attempt to match the luminous efficiency (http://www.cvrl.org/lumindex.htm) of the different LEDSs to make them equiluminant. Photometric luminance matching of the two lights is necessary if any differences in detection of “red” vs “green” are discussed.

It would be very difficult to photometrically match a “red” and “green” light without measuring an individual’s eyelid transmission, this seems to weaken any discussion in the context of anesthesia and closed eyelids.  I think the paper would be stronger without any discussion of the red/green lights.

Authors’ Response: We thank the reviewer for pointing out this important limitation. Indeed, we have calibrated the luminance intensity with which the LED goggles emitted green and red light at 1,000 lux before the experiment. This information has been added to the Experimental setup paragraph of the Materials and Methods section in the revised manuscript. We agree, however, that it remains unclear whether green and red light stimulation has truly resulted in different colour or different brightness stimulation through closed eyelids. This important aspect has been added to the Discussion of the revised manuscript. The suggested reference by Ando and Kripke was included. In addition, we attenuated the interpretation of predicting the cortical EEG response to green or red light stimulation in the revised manuscript. The respective part of the revised Discussion section reads as follows: “Although the light emitting diodes were calibrated to emit green and red light at the same luminance intensity, it is unclear whether colour stimulation through closed eyelids truly resulted in different colour stimulation of the eye. A study evaluating the illumination necessary to yield a visual threshold response to green and right light reported that the estimated light transmission through the eyelids was 0.3% for green and 5.6% for red light [13]. Therefore, it remains unclear if the cortical EEG responses corresponded to differences in colour or actually differences in brightness.” Furthermore, the conclusion paragraphs of the revised abstract and main text do not refer to the results of different colour stimulation any longer. Please see below.

Despite not taking into account brightness rather than just color of their light stimuli, I think there is a strength in this paper.  The strength of this paper is its use of the machine learning algorithms to quickly differentiate responses in subjects. It is also worth noting that the X boosting algorithms differentiated better than neural networks. This would be most useful in clinical care when speed and efficiency are critical, for example in getting VEP data from infants or very young children. Other applications in cases of patients with severe trauma and might not be able to respond or attend to a stimulus for long periods of time.

Authors’ Response: Thank you very much for your comment. In the Discussion section of the revised manuscript, we paid more attention to highlight the potential benefits of our machine learning-based algorithm to allow for both rapid and continuous VEP analyses. We included examples as suggested by this reviewer. The respective part of the revised Discussion section reads as follows: “From a clinical perspective, this appears to be the most important result and strength of our study. In contrast to standard, intermittent VEP interpretation by either neuro-physiologists or other machine learning models [12], our algorithm allows for nearly continuous VEP analysis. Possible clinical scenarios in which rapid or continuous VEP interpretation would be helpful are for example: VEP analysis in infants or very young children, patients with severe trauma unable to respond or attend to a stimulus for long periods of time, as well as real-time VEP analysis during surgery.”

Furthermore, we highlighted the finding that X-gradient boosting algorithms performed better than neuronal networks in the revised manuscript. In order to do so we introduced our analysis to identify the best performing machine learning algorithms into the Materials and Methods section of the revised manuscript. The respective paragraph reads as follows: “In order to identify the most appropriate machine learning algorithm to answer the primary study question, we performed the classification task to predict the cortical EEG response to no light or light stimulation in all study participants using five different machine learning algorithms (X-gradient boosting, random forests, neuronal networks, adaBoost, general linear models). The algorithm exhibiting the highest AUCROC was subsequently used to answer the primary and secondary study questions. The median and 95% confidence intervals of the AUCROC of the different machine learning algorithms to predict the cortical EEG response to no light or light stimulation in all study participants were as follows: X-gradient boosting, 0.91 (0.9-0.91); neuronal networks, 0.89 (0.88-0.91); general linear models, 0.89 (0.88-0.9); random forests, 0.87 (0.86-0.88); adaBoost, 0.76 (0.75-0.78).”

In addition, the results were discussed in more detail in the revised Discussion section. The respective part of the revised Discussion section reads as follows: “However, unlike in the previous study, in which a single layer neuronal network was applied [3], we used the tree-based X-gradient boosting model to analyse the raw EEG data, as this algorithm showed the highest predictive value to answer the main classification task in this dataset. X-gradient boosting models are effective in analysing biosignals and allow to build simpler algorithms from complex raw signals than neuronal networks. Similarly, when comparing the predictive value and accuracy of machine learning algorithm in this to the ones reported in our previous study, the X-gradient boosting model showed better prediction and accuracy than the neuronal network when classifying the VEP response to green or red colour stimulation in one individual when the algorithm was trained on the remaining study population [3].”

Finally, the conclusion paragraphs of both the main text and the abstract were partly reworded accordingly and now read as follows: “In conclusion, machine learning algorithms could almost continuously and reliably differentiate between the cortical EEG responses to no light or light stimulation using VEPs in awake male and female volunteers when their eyes were closed. Our findings may open new possibilities for the use of VEPs in the clinical and intraoperative setting.”

Round 2

Reviewer 1 Report

Comments and Suggestions for Authors

The format of the references  is wrong, such as references 9. Please revise it carefully.

Author Response

Authors’ Response: The format of all references has been carefully checked and adjusted to the recommended reference style of the journal. Following our discussion with reviewer 2, reference 25 was deleted from the reference list.

Reviewer 2 Report

Comments and Suggestions for Authors

The authors addressed or answered most of my concerns and comments.

I disagree with the authors revision that argues that the ability to detect red and green lights with the method is "unclear".  The Ando and Kripkie paper now referenced makes it clear there is different transmission of light through the eyelid despite equal luminance for the red and green lights.  This would result in a confound with more red light than green light transmission. 

The paper would be stronger just omitting any reference to the red/green lights because no conclusion can be drawn using them because of the confound of lid transmission.  Another less desirable alternative is to keep the red/green stimuli but only as another example of using the machine learning algorithms to differentiate two stimuli that vary in brightness.  

Author Response

Authors’ Response: In the revised manuscript, we clearly stated that red or green light stimulation through closed eyelids resulted in light stimulation with different brightness as indicated by the reference by Ando and Kripke. According changes were made to the Title, Abstract, Patients and Methods, and Results of the revised manuscript. As suggested by this reviewer, we discussed the results as another example of using the machine learning algorithms to differentiate between the cortical responses to light stimulation with different brightness. The respective parts of the revised manuscript now read as follows:

Title:

“Continuous detection of subtle differences in visual evoked potentials applied to closed eyes in healthy volunteers”

Abstract, Background/Objectives:

“Background/Objectives: We defined the value of a machine learning algorithm to distinguish between the EEG response to no light or any light stimulation, and between light stimulation with different brightness in awake volunteers with closed eyelids.”

Abstract, Methods:

“Methods: X-gradient boosting models were used to classify the cortical response to visual stimulation (no light vs. light stimulation and two lights with different brightness).”

Abstract, Results:

“The machine learning algorithm was highly predictive and accurate in distinguishing between light stimulation with different brightness ([…]).”

Patients and Methods section, paragraph Study Goals:

“As the primary study goals, we aimed to define the value of a machine learning algorithm to distinguish (1) […], and (2) between the cortical EEG response to light stimulation with different brightness using red or green light stimulation in awake volunteers with closed eyelids.”

Results section, paragraph 2:

“Similarly, the machine learning algorithm was highly predictive and accurate in correctly classifying the cortical response to light stimulation with different brightness (green or red colour stimulation) ([…]) (Figure 3).”

Results section, legend of Figure 3:

“Figure 3. Violin plots of the area under the receiver operator characteristic curve (AUC ROC) and accuracy for the prediction model including the entire study population (all), individual participants (individual), and all participants except one (one out) following light stimulation with different brightness (green or red light stimulation).”

Results section, legend of Figure 4:

“Figure 4. Violin plots of the area under the receiver operator characteristic curve (AUC ROC) and accuracy for both prediction models including the entire study population in females and males following no or any light stimulation, as well as following light stimulation with different brightness (green or red light stimulation).”

Discussion section, paragraph 1:

“In this experimental study, machine learning algorithms achieved an excellent predictive value and a very high accuracy to detect VEPs in the EEG signal and to distinguish between the VEP responses to no light or light stimulation as well as to light stimulation with different brightness in awake volunteers with closed eyelids.”

Discussion section, paragraph 2:

“Similarly, when comparing the predictive value and accuracy of machine learning algorithm in this to the ones reported in our previous study, the X-gradient boosting model showed better prediction and accuracy than the neuronal network when classifying the VEP response to light stimulation with different brightness (green or red light stimulation) in one individual when the algorithm was trained on the remaining study population [3].”

Discussion section, paragraph 5:

“Second, the machine learning algorithms in this study could not only distinguish between no light stimulation or light stimulation but also reliably classify the cortical VEP response to light stimulation with different brightness. Although the light emitting diodes were calibrated to emit green and red light at the same luminance intensity, the eyelid serves as a colour filter as the estimated light transmission through the eyelid was found to be 0.3% for green light and 5.6% for red light [13]. Therefore, the cortical EEG responses detected in our study corresponded to light stimulation with different brightness and not stimulation with different colours.”

Discussion section, limitations paragraph:

“In addition, we used green and red light to simulate light stimulation with different brightness in our study. This was the result of pilot tests analysing which light stimulations resulted in cortical VEP responses most different from each other. We can, therefore, not exclude that using only green or only red light with different brightness or another colour with different brightness might have resulted in an even better classification ability of the machine learning algorithms.”

Round 3

Reviewer 2 Report

Comments and Suggestions for Authors

The authors sufficiently addressed my concerns.

The only other issue remaining is the article title:  I would suggest something like:

Continuous detection of stimulus brightness differences using visual evoked potentials in healthy, closed eyed volunteers

Author Response

The title has been changed as requested.